# An explainable artificial intelligence approach for predicting cardiovascular outcomes using electronic health records

Sergiusz Wesołowski[1☯], Gordon Lemmon[1☯], Edgar J. Hernandez[1], Alex Henrie[1], Thomas A. Miller[2], Derek Weyhrauch[2], Michael D. Puchalski[2], Bruce E. Bray[3,4], Rashmee U. Shah[3], Vikrant G. Deshmukh[5], Rebecca Delaney[6], H. Joseph Yost[7], Karen Eilbeck[6], Martin Tristani-Firouzi[2,8]*, Mark Yandell[1]*

**1** Department of Human Genetics and Utah Center for Genetic Discovery, University of Utah, Salt Lake City, UT, United States of America, **2** Division of Pediatric Cardiology, University of Utah School of Medicine, Salt Lake City, UT, United States of America, **3** Division of Cardiovascular Medicine, University of Utah School of Medicine, Salt Lake City, UT, United States of America, **4** University of Utah, Biomedical Informatics, Salt Lake City, UT 84108, United States of America, **5** University of Utah Health Care CMIO Office, Salt Lake City, UT, United States of America, **6** Department of Population Health Sciences, University of Utah, Salt Lake City, UT, United States of America, **7** Molecular Medicine Program, University of Utah, Salt Lake City, UT, United States of America, **8** Nora Eccles Harrison CVRTI, University of Utah School of Medicine, Salt Lake City, UT, United States of America

☯ These authors contributed equally to this work.
* Martin.Tristani@utah.edu (MTF); myandell@genetics.utah.edu (MY)

**Data Availability Statement:** We obtained medical records from the University of Utah and Primary

## Abstract

Understanding the conditionally-dependent clinical variables that drive cardiovascular health outcomes is a major challenge for precision medicine. Here, we deploy a recently developed massively scalable comorbidity discovery method called Poisson Binomial based Comorbidity discovery (PBC), to analyze Electronic Health Records (EHRs) from the University of Utah and Primary Children's Hospital (over 1.6 million patients and 77 million visits) for comorbid diagnoses, procedures, and medications. Using explainable Artificial Intelligence (AI) methodologies, we then tease apart the intertwined, conditionally-dependent impacts of comorbid conditions and demography upon cardiovascular health, focusing on the key areas of heart transplant, sinoatrial node dysfunction and various forms of congenital heart disease. The resulting *multimorbidity networks* make possible wide-ranging explorations of the comorbid and demographic landscapes surrounding these cardiovascular outcomes, and can be distributed as web-based tools for further community-based outcomes research. The ability to transform enormous collections of EHRs into compact, portable tools devoid of Protected Health Information solves many of the legal, technological, and data-scientific challenges associated with large-scale EHR analyses.

## Introduction

The application of data-science methods to electronic health record (EHR) databases promises a new, global perspective on human health, with widespread applications for outcomes

Children's Hospital under an IRB that waived consent (see ethics statement). We refer to this cross-institution extract as the Utah Data Resource. Because the aggregate is comprised of exact dates and other protected patient information, the data cannot be made publicly available. Information regarding how qualified researchers might apply for data access can be found here https://irb.utah.edu/about/contact/. However, All Probabilistic Graphical Models described in this paper are available through the web using the following link: https://pbc.genetics. utah.edu/lemmon2021/bayes/.

**Funding:** This research was supported by the AHA Children's Strategically Focused Research Network grant (17SFRN33630041) (https://professional. heart.org/en/research-programs/strategically- focused-research/strategically-focused-research- networks) and the Nora Eccles Treadwell Foundation. RD's effort was supported by the National Institutes of Health under Ruth L. Kirschstein National Research Service Award T32 HL007576 from the National Heart, Lung, and Blood Institute (https://grants.nih.gov/grants/oer. htm). GL was supported by NRSA training grant T32H757632 (https://researchtraining.nih.gov/ programs/training-grants/T32). SW was supported by NRSA training grant T32DK110966-04 (https:// researchtraining.nih.gov/programs/training-grants/ T32). The funders had no role in study design, data collection and analysis, decision to publish, or preparation of the manuscript.

**Competing interests:** I have read the journal's policy and the authors of this manuscript have the following competing interests: GL, VD, MY own shares in Backdrop Health, there are no financial ties regarding this research.

research and precision medicine initiatives. However, unmet technological challenges still exist [1–3][. One is the need for improved means for *ab initio* discovery of comorbid clinical variables in the context of confounding demographic variables at scale. Moreover, how best to tease apart the intertwined impacts of multiple comorbidities and demographic variables on patient health remains a daunting challenge [1, 3–9].

We used a massively-scalable comorbidity discovery method called Poisson Binomial based Comorbidity (PBC) discovery [10] to search Electronic Health Records (EHRs) from the University of Utah and Primary Children's Hospital for comorbid diagnoses, procedures, and medications. In this context, we refer to co-occurring medical diagnoses, procedures and medications using the single blanket term, comorbidity. PBC can also discover temporal relationships and quantify transition rates between various comorbidities. The result is a disease network, devoid of Protected Health Information (PHI), that is well-suited for powering downstream outcomes research.

Although comorbidity discovery is a necessary first step towards enabling outcomes research, it is not an end in itself. Comorbidities do not exist as isolated pairs, rather they combine to create a complex web of influence on any given outcome. While PBC is powered to discover that web, harnessing it for outcomes research requires a separate computational machinery, one capable of calculating the joint contributions of multiple, conditionally dependent variables on an outcome, so called *multimorbidity* calculations [1,3,11–13]. Moreover, because researchers seek not merely to predict outcomes, but also to measure the contributions of factors driving them, 'explainable' solutions [14–22], rather than black box approaches are required. We have adapted Probabilistic Graphical Models (PGMs) [2,22–27] to address these needs.

PGMs are well suited for outcomes research. Contrary to other methods, e.g. generalized linear models (with or without mixed effects), PGMs are capable of: (1) discovering and modelling any number of multilevel dependencies between variables, (2) capturing non-additive or non-multiplicative interactions, and (3) their application does not require excluding nor imputing missing data [28]. Moreover, PGMs model the full joint probability function governing relationships in the data, and thus do not necessitate a dichotomy between response and input variables. Rather, PGMs are capable of answering a prediction query for any variables conditioned on any set of inputs included in the model.

Using these computational technologies, we mined the EHRs of over 1.6 million University of Utah and Primary Children's Hospital patients, including over 500,000 mother-child pairs, for comorbid diagnoses, procedures, medications, and lab tests driving diverse cardiovascular health outcomes, focusing on three areas: heart transplant, sinoatrial node dysfunction, and congenital heart disease. Our results illuminate the comorbid and demographic landscapes surrounding these key cardiovascular outcomes in the US intermountain west, and demonstrate how our approach can inform health care disparities with precise, quantitative results in the context of a specific health care system.

## Results

### PBC is well powered for discovery of cardiovascular comorbidities

Table 1 demonstrates the utility of the PBC [10] approach for discovery, by comparing the power of PBC versus a standard stratification approach (followed by $\chi^2$) to detect the well documented comorbid relationship between atrial fibrillation (AF) and acute cerebrovascular disease (stroke) [29,30]. Table 1 provides a power analysis as a function of corpus size and number of demographic variables. The effects of stratifying the data for $\chi^2$ analysis, versus adding them to the PBC calculation, can be observed as one proceeds down the table columns.

**Table 1.  PBC is well powered for comorbidity discovery on demographically complex datasets, unlike stratification.**

| Atrial Fibrillation and Acute Cerebrovascular Disease | | | | | |
|---|---|---|---|---|---|
| Features | PBC p-value | | | χ2 p-value | | |
| | N = 1,538,059 | N = 95,407 | N = 9,525 | N = 1,538,05 | N = 95,407 | N = 9,525 |
| no features | 1e-31020 | 1e-1715 | 1e-203 | 1e-31020 | 1e-1715 | 1e-203 |
| +sex | 1e-31017 | 1e-1955 | 1e-215 | 1e-16657 | 1e-1125 | 1e-147 |
| +age | 1e-25448 | 1e-1589 | 1e-200 | 1e-1304 | 1e-88.3 | 1e-13.1 |
| +ancestry | 1e-14381 | 1e-628 | 1e-73.1 | 1e-15.72 | 1 | 1 |
| +ethnicity | 1e-11357 | 1e-806 | 1e-110 | 1e-12.25 | 1 | 1 |
| +insurance | 1e-11533 | 1e-771 | 1e-83 | 1e-2.68 | 1 | 1 |
| +span | 1e-11325 | 1e-698 | 1e-84.1 | 1e-1.75 | 1 | 1 |

Progressively smaller random samples were drawn from the Utah EHR corpus, such that each cohort is a subset of this larger precursor. N = the number of subjects in each cohort under consideration. Cells in the table contain p-values for the association between Atrial Fibrillation and Acute Cerebrovascular Disease (stroke), as calculated by PBC or χ2 (for stratification). P-values less than the Bonferroni corrected alpha (1e-9.5) are shown in light blue, while cells that do not pass the significance threshold are red. Stratum filters apply to the features' column, row by row as follows: no filters, female, 50–59 years of age, white, non-Hispanic, commercial insurance, minimum of 2 years of medical history.

Results for three different starting cohort sizes are shown. Note how stratification lowers the strength of p-values as a function of the size of the stratum. This effect is exacerbated when more than a few potentially confounding variables are controlled for, and stratification quickly results in cohorts that are too small for discovery activities, as the comorbidities fail to achieve statistical significance. For example, using a starting corpus of 9,525 records, stratification followed by χ² analysis fails to detect the *well-known* comorbid relationship between AF and Stroke for female patients aged 50–59 when white ancestry is included in the stratum description. By contrast, the PBC approach maintains power across all comparisons. For more on these points, see [10].

## Comorbidities of heart transplant

We evaluated every pairwise combination of diagnoses, procedures, and medications mentioned in our EHR corpus for comorbid associations, using PBC [10] to adjust on a patient-by-patient basis for the potentially confounding demographic variables shown in **Fig 1**. **Fig 2A** summarizes the results of this computation as a patient disease network. The network provides a visual overview of the entire EHR corpus, wherein every node (state) is a diagnosis, procedure, or medication, and edges denote Bonferroni significant comorbid relations between terms. Given a node of interest, heart transplant, for example, its comorbid diagnoses and associated procedures and medications can be recovered by following edges to that node back to their terms.

The transition probabilities associated with each edge provide means to calculate the pairwise contributions of each term to the outcome's observed (marginal) frequency in the EHR corpus. This provides a way to intuit an outcome's comorbidity landscape, and calculate the expected flux of patients through that region of the network. These patient 'trajectories' provide a framework for cost prediction and service allocation activities. For example, the trajectory for adult heart transplant (**2B**) tracks the time course of diagnoses, procedures and medication use preceding and following heart transplantation. Thus, one can follow the trajectory of ischemic heart disease, flowing through the diagnosis of heart failure, cardiogenic shock, administration of the vasoactive medication milrinone, and culminating in heart transplantation with subsequent downstream complications. Crucially, this methodology provides precise measures of patient flux between these nodes.

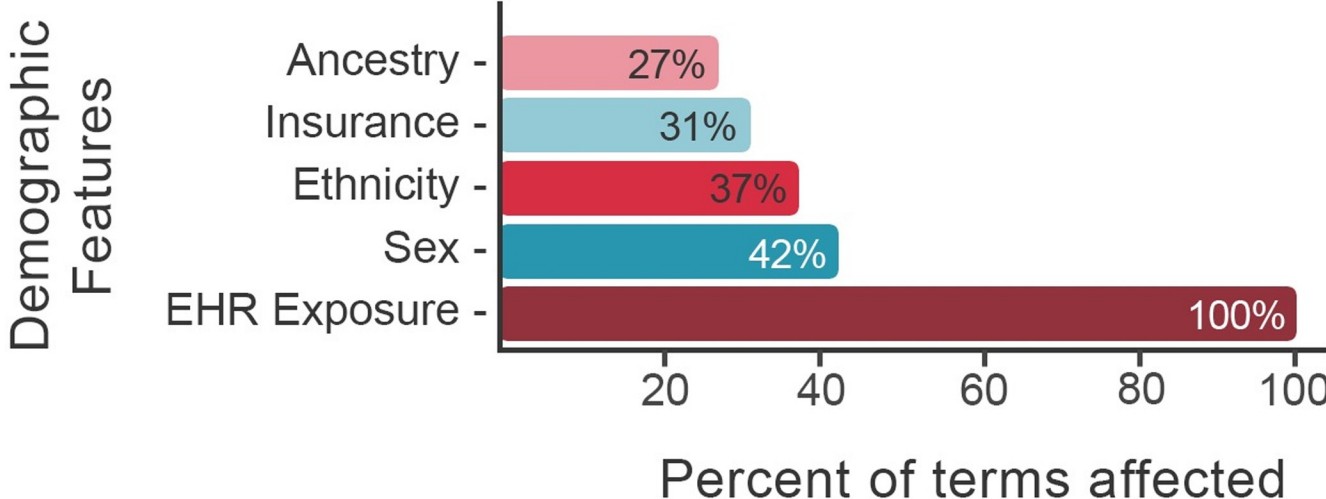

**Fig 1. Percent of medical terms influenced by various demographic features.** Demographic variables used in the comorbidity discovery process are displayed on the y-axis. The percent of all diagnoses, procedures, and medications influenced by a given demographic feature is displayed on the x-axis. For example, sex influences 42.2% percent of diagnoses, procedures, and medications in the Utah EHR corpus; ancestry influences 27.4% and EHR exposure 100%. EHR exposure includes subject age, length of medical record history, number of visits. See article [10] for details. Features were selected using L1 regularization.

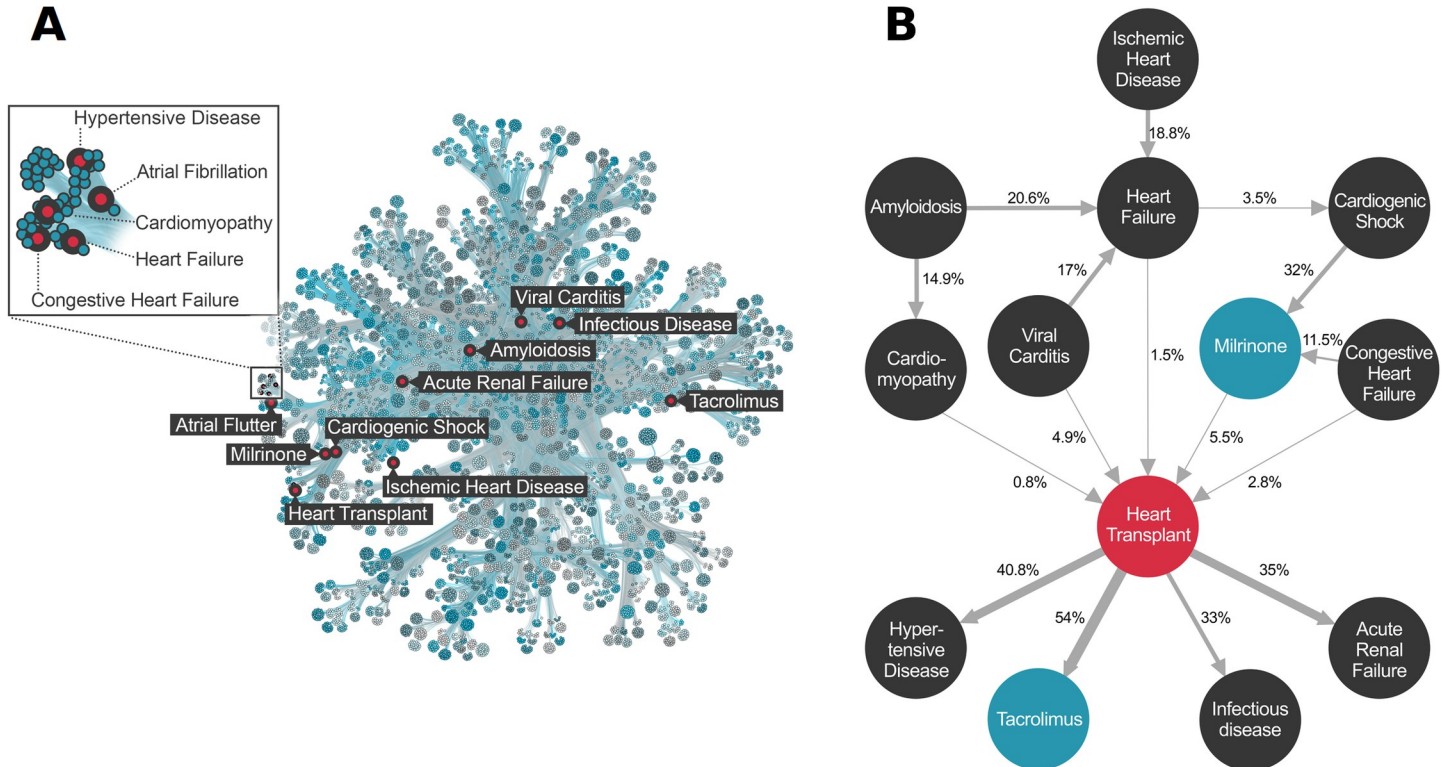

**Fig 2. Patient Disease Network for the Utah Data Resource. Panel A.** Graphical representation of the Patient Disease Network. 39,055 ICD 10 diagnosis codes, 5,716 CPT procedure codes, and 1,764 RxNorm medication codes comprising 50 million comorbidities are represented by the map. To render the patient disease network more readily interpretable, we utilized Minimum Description Length clustering, so that nodes with similar comorbidity patterns lay near to one another in the network. The comorbidities of Heart Transplant are labeled red for reference purposes. See Methods for details. **Panel B.** Term trajectory for Adult Heart Transplant. Nodes represent diagnosis (black), procedures (red), and medications (blue). Edges are temporally ordered comorbidities (Bonferroni alpha = 10E-9.5), arrows denote direction. Edges are labeled with transition probabilities (e.g. patient flux). For example, an adult patient with viral myocarditis has a 17% chance of developing a heart failure diagnosis, and a 4.9% chance of undergoing heart transplantation. See Methods for additional details and **S5 Table** for code references for the highlighted terms.

## Multimorbidity network for heart transplant supports conditional outcome risk calculations

Although trajectories provide intuitive and useful overviews of the comorbidity landscape, effective outcomes research requires calculating the *joint* contributions of conditionally dependent multimorbid terms on an outcome. We leverage Probabilistic Graphical Models as an explainable AI solution for this computationally intensive task. **Fig 3A** illustrates a multimorbidity network derived from a temporalized Probabilistic Graphical Model for the predisposing comorbidities of adult heart transplant presented in **Fig 2B.** Because the edges in a multimorbidity network denote conditional dependencies between terms, rather than transition probabilities, the multimorbidity network's topology is necessarily different from the trajectory topology shown in **Fig 2B**. The PGM provides easy means to calculate outcomes risk for any combination of variables in it. For example, a prior diagnosis of cardiomyopathy (non-ischemic) increases the risk of heart transplantation 86±35 fold, whereas a diagnosis of viral myocarditis confers a 59±21 fold increase in risk. The strongest single variable for heart transplant risk is the use of the vasoactive medication milrinone, which increases risk 175±30 fold. Note that we are not suggesting milrinone causes heart transplant—rather that the prescription of milrinone in a patient's medical record is a powerful predictor of future heart transplant.

The utility of PGMs for outcomes research is best illustrated by their application to problems of complex multimorbid outcomes analyses, where conditional dependencies of these variables interact to further modulate risk for the outcome under study. For example, we can explore the role of heart disease etiology on transplant risk in the context of milrinone infusion. Thus, a cardiomyopathy patient requiring milrinone has a 407±101 fold increased risk for heart transplant. Likewise, a patient with viral myocarditis requiring milrinone therapy has a 346±93 fold increased risk for heart transplant; while milrinone use in a patient with ischemic heart disease confers a 205±28 fold increased risk of heart transplant. Moreover, while both cardiomyopathy and ischemic heart disease have similar increased risks for heart transplant in isolation (86±35 fold and 64±14 fold, respectively), cardiomyopathy patients who require milrinone therapy are at far greater risk for heart transplant than patients with ischemic heart disease requiring milrinone. Additional conditional queries conducted with the PGM are presented in **Fig 3A.** This list is by no means exhaustive—the PGM is capable of answering an astonishing number of queries—$3^{25}$ to be precise. We encourage the reader to explore these by following the link to the corresponding web application https://pbc.genetics.utah.edu/lemmon2021/bayes. In this context, the explainable nature of PGMs lays the foundation for massively parallel testing of novel hypotheses between multiple, complex clinical variables of interest.

The comorbidity landscape for pediatric heart transplant is dramatically different from that of adults, as it includes a large contribution from congenital heart defects (CHD) and palliative surgical procedures. **Fig 3B** presents a multimorbidity network for 13 common CHD terms defined by echocardiogram and identified by PBC as comorbid with pediatric heart transplant. A prior diagnosis of dilated cardiomyopathy (DCM), defined as genetic/idiopathic DCM, increases a child's risk for heart transplant 102.2±33.6-fold, over the marginal probability of transplant. Among single ventricle forms of CHD, patients with hypoplastic left heart syndrome (HLHS) are at the greatest risk for heart transplant (56.8±17.8-fold), as compared to tricuspid atresia (17.1±11.8-fold) or laterality defects (25.8-fold ± 8.5). Again, the utility of PGMs for complex multimorbid analyses is highlighted by the ability to calculate the additional risk for heart transplant in a child with a laterality defect, if that child also requires the Norwood surgery (51.3±10.5-fold).

| CLINICAL VARIABLES | RISK | FOLD CHANGE | BAYESIAN NETWORK |
|---|---|---|---|
| Milrinone | 0.054 +/-0.009 | 175 +/- 30 | |
| Cardiomyopathy | 0.026 +/- 0.011 | 86 +/- 35 | |
| Congestive Heart Failure | 0.025 +/- 0.004 | 81 +/- 14 | |
| Cardiogenic Shock | 0.05 +/- 0.03 | 69 +/- 7 | |
| Ischemic Heart Disease | 0.019 +/- 0.004 | 64 +/- 14 | |
| Viral Carditis | 0.018 +/- 0.006 | 59 +/- 21 | |
| Heart Failure | 0.03 +/- 0.02 | 48 +/- 12 | |
| Amyloidosis | 0.010 +/- 0.002 | 34 +/- 7 | |
| Cardiomyopathy + Milrinone | 0.125 +/- 0.036 | 407 +/- 101 | |
| Viral Carditis + Milrinone | 0.106 +/- 0.028 | 346 +/- 93 | |
| Ischemic Heart Disease + Milrinone | 0.063 +/- 0.010 | 205 +/- 28 | |

A

| CLINICAL VARIABLES | RISK | FOLD CHANGE | BAYESIAN NETWORK |
|---|---|---|---|
| DCM | 0.273 +/-0.122 | 102.2 +/- 33.6 | |
| HLHS | 0.147 +/- 0.047 | 56.8 +/- 17.8 | |
| Norwood | 0.096 +/- 0.031 | 37.1 +/- 12.8 | |
| Glenn | 0.089 +/- 0.055 | 34.3 +/- 21.8 | |
| Fontan | 0.068 +/- 0.041 | 26.1 +/- 16.1 | |
| Laterality Defects | 0.067 +/- 0.022 | 25.8 +/- 8.5 | |
| Tricuspid Atresia | 0.045 +/- 0.031 | 17.1 +/- 11.8 | |
| AVSD | 0.014 +/- 0.014 | 5.2 +/- 5.0 | |
| Coarctation | 0.006 +/- 0.002 | 2.3 +/- 0.8 | |
| ASD | 0.005 +/- 0.002 | 1.9 +/- 0.7 | |
| VSD | 0.003 +/- 0.003 | 1.3 +/- 1.2 | |
| TOF | 0.003 +/- 0.001 | 1.0 +/- 0.1 | |
| BAV | 0.002 +/- 0.000 | 0.7 +/- 0.1 | |
| Norwood + Laterality Defects | 0.124 +/- 0.028 | 51.3 +/- 10.5 | |

B

**Fig 3. Multimorbidity Landscape of Heart Transplant. Panel A.** PGM for Adult Transplant. N = 1.6 million individuals. The clinical variables were chosen based on Bonferroni-corrected ICD10 and RXnorm billing codes significantly associated (preceding) with heart transplant. Each node represents a diagnosis, procedure, or medication code and each edge represents a conditional dependence between nodes. For detailed description of the clinical variables, please refer to S5 Table. **Panel B.** PGM for Pediatric Transplant. N = 26,458 individuals. Clinical variable terms represent terms in the Primary Children's Hospital echocardiographic database or CCS billing codes when available. For detailed description of the clinical variables, please refer to the S5 Table. DCM: Dilated cardiomyopathy; Norwood: Norwood surgery; HLHS: hypoplastic left heart syndrome; Glenn: Glenn surgery; Fontan: Fontan surgery; AVSD: atrioventricular septal defect; ASD: Atrial septal defect; BAV: Bicuspid aortic valve; Coarctation: Coarctation of the aorta; VSD: Ventricular septal defect. Heart Transplant is highlighted in orange. For A and B, the target node (heart transplant) is colored red and nodes with direct connections to the target (ie, within the Markov blanket) are circled red. Values in Tables represent mean ± STD.

## Multimorbidity network for sinoatrial node dysfunction supports multimorbidity risk calculations for a range of clinical and demographic health predictors

Fig 4A extends the investigations to include the impacts of these same pediatric heart surgeries in the context of various CHD phenotypes on a different clinical outcome, sinoatrial node dysfunction (SND). The Fontan surgery dominates the landscape of pediatric SND, increasing the risk 19.6±6.4-fold over the marginal probability of SND. Moreover, Fontan surgery is the only clinical variable with a direct connection to SND; the other clinical variables connect indirectly to SND via the Fontan node. Thus, the relative risk of SND for specific forms of single ventricle CHD (HLHS, tricuspid atresia, unbalanced AVSD) following the Fontan surgery are similar (Fig 4), indicating that the Fontan surgery itself is the primary indicator of future SND, rather than the underlying form of CHD that required the procedure. Collectively, the preceding analyses demonstrate how multiple nets can be used in tandem to address complex multimorbidity outcomes questions.

Multimorbidity networks also provide powerful means to investigate the impacts of various demographic factors upon outcomes. The net in Fig 4B models the multimorbid landscape surrounding SND in adult patients. As SND and AF are both risk factors for each other [31], we temporalized the network (see Methods) to analyze clinical variables that precede SND. The ancestry and ethnicity nodes enable explorations of demographic impacts upon SND and its comorbidities. Thus, in the University of Utah Hospital system, a Hispanic patient with AF has a 61±6 fold increased risk of SND, compared to 30±1 fold risk for white ancestry and 40±7 fold risk for African Americans. These results underscore the potential of our approach to inform ethnic/racial health care disparities with precise, quantitative results, and in the context of a specific health care system. Moreover, these findings illustrate how our approach can empower these discussions despite demographic skews in the underlying EHR corpus (see S2 and S3 Tables); an important finding for the Utah health system.

## Multimorbidities of congenital malformations augmented by maternal health data

The impact of maternal health on health outcomes in the child is an area of intense investigation. The Multimorbidity network shown in Fig 5A places a child's risk for congenital malformations in the context of a maternal diagnosis of pregnancy-induced hypertension (HTN-PREG) during that pregnancy, leveraging outcomes data for over 130,000 births at the University of Utah Hospital system over the last 15 years. HTN-PREG elevates the risk of cardiac and circulatory congenital anomalies 1.83±0.03-fold, an effect not due to maternal age differences (S1 Fig). The multimorbidity network also illuminates the strong dependencies between clinical variables and allows for quantitative assessments of risk. For example, a diagnosis of Down Syndrome is associated with a 25.9±0.8-fold increased risk for a congenital cardiac anomaly (S4A Table). Moreover, a child with a congenital cardiac anomaly is *a priori* 9.2±0.9-fold more likely to have a nervous system anomaly than baseline (S4B Table). The impact of maternal health on a child's

| CLINICAL VARIABLES | RISK | FOLD CHANGE | BAYESIAN NETWORK |
|---|---|---|---|
| Fontan | 0.10+/-0.03 | 19.6 +/- 6.4 | |
| Tricuspid Atresia | 0.07 +/- 0.03 | 12.8 +/- 6.2 | |
| Heart Transplant | 0.06 +/- 0.07 | 11.5 +/- 12.8 | |
| HLHS | 0.05 +/- 0.03 | 8.9 +/- 4.9 | |
| Norwood | 0.04 +/- 0.02 | 8.3 +/- 4.2 | |
| AVSD unbalanced | 0.03 +/- 0.03 | 6.4 +/- 5.4 | |
| Laterality Defects | 0.03 +/- 0.02 | 6.1 +/- 3.5 | |
| dTGA | 0.03 +/- 0.02 | 6.0 +/- 3.5 | |
| RV fxn decreased | 0.02 +/- 0.01 | 4.0 +/- 2.3 | |
| TR | 0.02 +/- 0.01 | 3.6 +/- 1.7 | |
| BAV | 0.01 +/- 0.00 | 1.0 +/- 0.0 | |
| HLHS + Fontan | 0.01 +/- 0.04 | 18.8 +/- 7.6 | |
| AVSD unbalanced + Fontan | 0.10 +/- 0.04 | 20.0 +/- 8.0 | |
| Tricuspid Atresia + Fontan | 0.11 +/- 0.04 | 21.2 +/- 6.9 | |

| CLINICAL VARIABLES | RISK | FOLD CHANGE | BAYESIAN NETWORK |
|---|---|---|---|
| Atrial Fibrillation | 0.14+/-0.01 | 31.0 +/- 1.1 | |
| Atrial Flutter | 0.08 +/- 0.00 | 17.6 +/- 0.7 | |
| Tachycardia | 0.06 +/- 0.00 | 12.3 +/- 0.6 | |
| Viral Carditis | 0.04 +/- 0.00 | 9.0 +/- 0.3 | |
| DCM | 0.04 +/- 0.00 | 9.0 +/- 0.3 | |
| Hypertension | 0.02 +/- 0.00 | 3.5 +/- 0.1 | |
| Obesity | 0.01 +/- 0.00 | 1.7 +/- 0.0 | |
| AS | 0.01 +/- 0.00 | 1.3 +/- 0.2 | |
| Coarctation | 0.01 +/- 0.00 | 0.9+/- 0.1 | |
| Female | 0.01 +/- 0.00 | 1.0 +/- 0.0 | |
| Atrial Fibrillation + Hispanic/Latino | 0.28 +/- 0.03 | 61 +/- 6 | |
| Atrial Fibrillation + Caucasian | 0.14 +/- 0.01 | 30 +/- 1 | |
| Atrial Fibrillation + African American | 0.19 +/- 0.03 | 40 +/- 7 | |

**Fig 4. Multimorbidity Landscape of Sinoatrial Node Dysfunction (SND).** Each node represents a diagnosis or procedure, each edge represents a conditional dependence between nodes. For detailed description of the clinical variables, please refer to **S5 Table**. **Panel A.** Pediatric SND. N = 26,458 individuals. Clinical variable terms represent terms in the Primary Children's Hospital echocardiographic database or CCS billing codes when available. Fontan: Fontan surgery; HLHS: hypoplastic left heart syndrome; Norwood: Norwood surgery; dTGA: d-transposition of the great arteries; RV fxn: right ventricular function; TR: > = moderate tricuspid regurgitation; BAV: bicuspid aortic valve. **Panel B.** Adult SND. N = 1.6 million individuals. Clinical variable terms represent CCS billing codes. Ancestry: Western European, African American, or Other; Ethnicity: Hispanic, or non-Hispanic. DCM: Dilated cardiomyopathy; AS: Aortic stenosis; Coarctation: Coarctation of the aorta. SND is highlighted in red in both panels. The target node (SND) is colored red and nodes with direct connections to the target (ie, within the Markov blanket) are circled red. Values in Tables represent mean ± STD.

risk of CHD is further explored in **Fig 5B**. Our ability to seamlessly combine and compute upon maternal/child EHR data highlights the extensibility of our approach to study health outcomes across generations in order to define the impacts of maternal health on childhood outcomes.

## Web-based outcomes calculators

We repackaged the multimorbidity networks as stand-alone web-based outcomes calculators. This allows users to interact with a multimorbidity network as an 'app', whereby they can use slider buttons to toggle values of its states and to select an outcome of interest. These web-apps are available here: https://pbc.genetics.utah.edu/lemmon2021/bayes/bayes.

## Methods

### Ethics statement

Human subjects approval for this study was obtained following review by the University of Utah Institutional Review Board, IRB_00095807 under a waiver of consent and authorization. Patient data was not anonymized prior to the start of the study. All authors completed Human Subjects research requirements.

### Utah data resource

The University of Utah maintains an Enterprise Data Warehouse (EDW)–a central storage and search facility for all clinical data collected from all affiliated University hospitals and clinics across the Intermountain West. SQL queries were used to aggregate data from various tables and collect the following information: (1) gender, ancestry, ethnicity, and age for each patient; (2) list of patient visits, along with visit dates, and medical terms associated with each visit, including diagnostic codes, procedure codes, and medications ordered. ICD9 and ICD10 diagnosis codes consist of 18,000 and 142,000 codes respectively, while procedural codes (CPT) include around 10,000 codes. In all, we collected records for 1.6 million patients, 21 million visits and 166 million diagnosis (DX), procedure (PX) and medication (RX) codes. See **S1–S5 Tables** for additional details.

We combined these data with the Primary Children's Hospital's database of echocardiographic variables (diagnoses, ventricular function, valve gradients, chamber/vessel sizes, etc.) dating back to 2006 for 65,618 probands, 44,254 of which also appear longitudinally in the University's EDW. These data contain 529,317 mother-child pairs with EHR data, 14,155 of which include a child with echo data, allowing us to study maternal contributions to congenital heart disease (CHD). Collectively, these data comprise the Utah Data Resource (UDR). For the purposes of computation, custom encryption is applied to the UDR to produce data free of protected health information (PHI) and unintelligible without its cyphers. We can then generate statistics on this PHI free data in a variety of compute environments, decrypting the results on PHI approved machines.

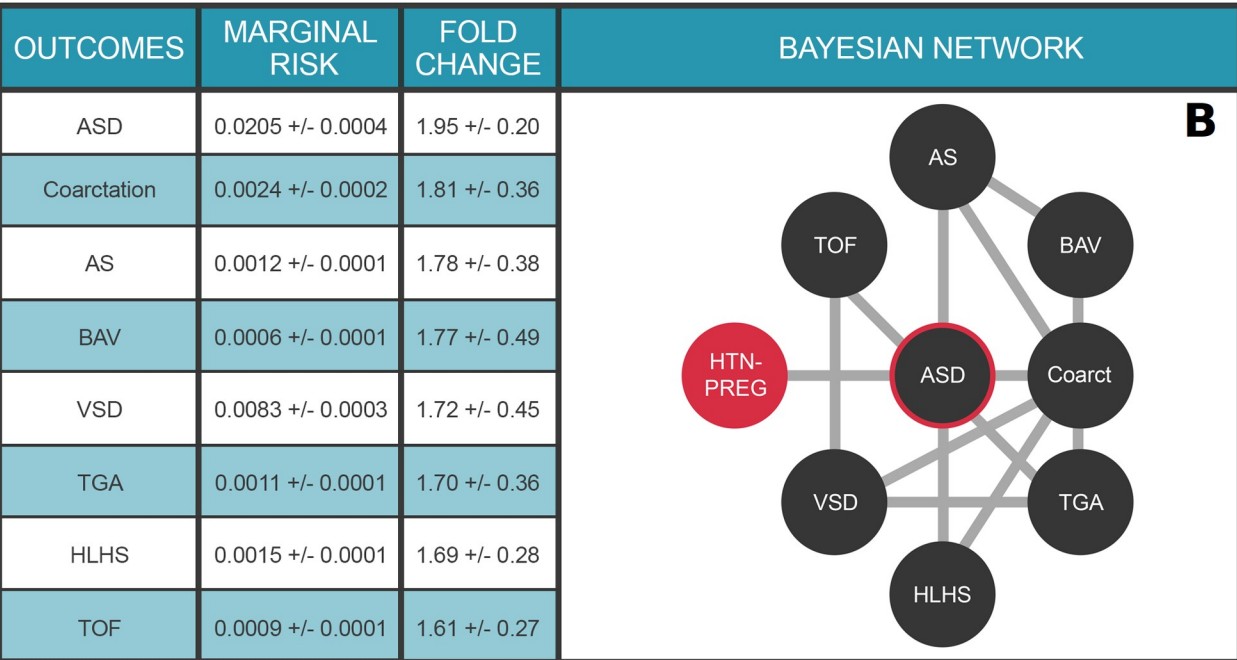

| OUTCOMES | MARGINAL RISK | FOLD CHANGE | BAYESIAN NETWORK |
|---|---|---|---|
| Skeletal | 0.0268 +/- 0.0002 | 1.89 +/- 0.03 | **A** |
| Cardiac | 0.0277 +/- 0.0002 | 1.83 +/- 0.03 | |
| Skin | 0.0160 +/- 0.0001 | 1.76 +/- 0.03 | |
| Downs | 0.0023 +/- 0.0001 | 1.65 +/- 0.07 | |
| Diaphragm | 0.0009 +/- 0.0000 | 1.43 +/- 0.05 | |
| Nervous | 0.0120 +/- 0.0004 | 1.37 +/- 0.15 | |
| Genito-Urinary | 0.0288 +/- 0.0002 | 1.34 +/- 0.02 | |
| Cleft Lip | 0.0014 +/- 0.0001 | 1.19 +/- 0.06 | |
| Digestive | 0.0113 +/- 0.0001 | 1.16 +/- 0.01 | |
| Eye | 0.0096 +/- 0.0001 | 1.13 +/- 0.01 | |

| OUTCOMES | MARGINAL RISK | FOLD CHANGE | BAYESIAN NETWORK |
|---|---|---|---|
| ASD | 0.0205 +/- 0.0004 | 1.95 +/- 0.20 | **B** |
| Coarctation | 0.0024 +/- 0.0002 | 1.81 +/- 0.36 | |
| AS | 0.0012 +/- 0.0001 | 1.78 +/- 0.38 | |
| BAV | 0.0006 +/- 0.0001 | 1.77 +/- 0.49 | |
| VSD | 0.0083 +/- 0.0003 | 1.72 +/- 0.45 | |
| TGA | 0.0011 +/- 0.0001 | 1.70 +/- 0.36 | |
| HLHS | 0.0015 +/- 0.0001 | 1.69 +/- 0.28 | |
| TOF | 0.0009 +/- 0.0001 | 1.61 +/- 0.27 | |

**Fig 5. Impact of maternal health on congenital anomalies in the child. Panel A.** Multimorbidity landscape for child's risk for congenital malformations in the context of pregnancy-induced hypertension. N = 125,014 mothers. Clinical variable terms represent CCS billing codes present in the EHR database. Maternal diagnosis is highlighted in orange; HTN-Preg: Maternal diagnosis of hypertension complicating pregnancy (aka, pregnancy-induced hypertension); Diaphragm: Diaphragmatic congenital abnormalities; Genito-Urinary: Genito-Urinary congenital abnormalities; Cardiac: Cardiac and Circulatory congenital abnormalities; Skeletal: Skeletal congenital abnormalities; Down: Trisomy 21; Digestive:

Congenital abnormalities of the gastrointestinal tract; Nervous: Nervous system congenital abnormalities; Eye: Congenital abnormalities of the Eye; CleftLip: Cleft lip. **Panel B.** Multimorbidity landscape for child's risk of congenital heart defects in the context of pregnancy-induced hypertension. N = 125,014 mothers. ASD, atrial septal defect; VSD, ventricular septal defect, HLHS, hypoplastic left heart syndrome; Coarctation, coarctation of the aorta; TOF, tetralogy of fallot; BAV, bicuspid aortic valve. For detailed description of the clinical variables, please refer to **S5 Table**. The target node (HTN-PREG) is colored red and nodes with direct connections to the target (ie, within the Markov blanket) are circled red. Values in Tables represent mean ± standard deviation.

In this analysis, a patient's diagnoses are inferred via billing codes. Thus, the investigations and risk calculations presented herein reflect medical practice within the University of Utah Hospital network and Primary Children's Healthcare. How closely they approximate underlying universal ('true') risks is still unknown. Moving forward, we note that the methods described below provide powerful means for large-scale cross institutional comparisons aimed at discovering differences in medical practice and billing trends.

## Patient disease network

We used a Poisson Binomial based methodology called PBC [10] to discover comorbidities within our EHR corpus. Standard methods such as stratification seek to control for confounding variables through 'stratifying' by age and gender (for instance) and calculating comorbidity statistics for each strata, under the restrictive assumption a every patient in a stratum has the same probability of manifesting each morbidity. However this approach fails to scale, since the use of many confounding variables leads to strata too small to detect a statistical significance comorbidity. In contrast, PBC models the effects of age, gender, race, ethnicity, insurance type, and the length and density of each patient's medical record. These input features are used to determine per-patient probabilities for each medical term, using a Poisson binomial test. The result is much greater statistical power [10].

PBC was used to find significant connections among every possible combination of ICD diagnoses, procedures, and RxNorm medication terms, thereby creating a patient disease network [10]. Patient disease network is a term borrowed from Capobianco et. al[3] and denotes a network comprising all significant connections among diagnoses, procedures and medications (Bonferroni p-value cutoff 10E-9.48). We only considered terms appearing in at least 15 patients. This filter reduced the number of unique terms to 39,055 ICD10 diagnosis codes, 5,716 CPT procedure codes, and 1,764 RxNorm medication codes. We used Minimum Description Length clustering [32] to visualize the data, so that nodes with similar combinations of edges would lay near one another in the network. We also determined the patient flux between every pair of nodes. The result is shown in **Fig 2A**, which provides a visual representation of our patient disease network for the entire EHR corpus.

In keeping with previous work [13,33–36] on patient disease networks, we refer to a sub-portion of the network, focused on a single outcome as a trajectory, or term trajectory. **Fig 2B** shows a trajectory for adult heart transplant. Trajectories provide means to display additional features of the network, such as transition probabilities (which correspond to patient flux between nodes), and the marginal frequencies of outcomes and comorbid terms within the EHR corpus. Collectively, this information allows for better intuition of the disease landscape surrounding an outcome. The trajectory is also a useful starting point for cost and service allocation calculations.

## Multimorbidity networks

While trajectories describe transition probabilities between two comorbid terms, they provide no means to determine the combined effects of multiple comorbid diagnoses, and associated clinical procedures and medications upon an outcome. We have employed Probabilistic

Graphical Models (PGMs) to overcome this limitation. We learned the structures of the PGMs using the python3 package "pomegranate" [28], which provides a Bayesian Information Criterion (BIC)-based DP-A* exact structure search algorithm [37,38,46]. The exact search algorithm explores the entire applicable space of conditional dependencies in order to discover the optimal network structure for the data. Parameter learning for this optimal network is accomplished using the loopy belief propagation algorithm [39]. We use the same package for our inference and multimorbidity risk calculations. The visual interpretation was designed using the graph_tool [40] Python3 package and D3.js Java library.

For each Probabilistic Graphical Model, a maximum of 25 comorbid features were selected using PBC and validated by experts in the medical field (TAM, DW, MDP, BEB, RUS, MTF). Features that were judged to be of clinical relevance, importance or interest for the field under study were selected and used as inputs to learn the PGM structure and infer risk. These selected features became the inputs used to learn the PGM structure and infer risk. The patient's features were described in a categorical data format, (e.g. indicating the ancestry, ethnicity, or insurance type) or "present/absent" binary variables in case of medical diagnoses and procedures. A continuous feature (e.g. age, BMI, blood pressure) were discretized based on established clinical thresholds. Because the PGMs only present the facts about the data, PGMs themselves cannot discover or infer the temporal order of the events (unless specified as a Dynamic PGM). To overcome this issue, for our temporalized PGMs we have imposed the order (discovered using PBC; see [10] for additional details.) on the EHR extraction process prior to learning the Probabilistic Graphical Model structure. When trained on temporalized data, PGMs are forced to learn temporal conditional probabilities. Missing data are handled inherently by the Probabilistic Graphical Model structure learning process. That is, no patients were excluded due to missing data and no missing data was imputed. The resulting temporalized structures we call multimorbidity networks.

Probabilistic Graphical Models represent conditional dependencies in the dataset as a directed acyclic graph (DAG); however, it is important not to confuse directionality with causality or temporal ordering. In keeping with best practice, the multimorbidity networks are visualized in their undirected, moralized form, in which every node is connected to its Markov blanket. A single constructed multimorbidity Network provides an inference engine capable of answering $O(3^n)$ personalized conditional risk queries, where n denotes the number of features describing a patient's condition, and the base of the exponent is 3, because in case of binary health records data there are three states for each node that can be specified: present, absent, or status unknown.

### Confidence values

Risk estimates derived from Probabilistic Graphical Models are maximum likelihood estimates given the optimal structure under the BIC and an assumed uniform prior probability of any distinct EHR. To obtain standard deviation values for these estimates, we created 100 nets in parallel [41] from bootstrap replicates of the same data used to create **Figs 3, 4 and 5**. We then queried the resulting replicate nets, and calculated standard deviations of risks of outcomes of interest.

### Discussion

The ability to model dependencies among multiple risk factors is crucial for meaningful outcomes research. Unfortunately, traditional techniques, such as logistic regression, have limited ability to capture so-called 'conditional dependencies' between variables, which are the heart and soul of multimorbid analyses. Although mixture and generalized linear models with

mixed effects can (in principle) overcome this weakness, these techniques are limited because a new model must be designed for every question. Neural nets provide one possible alternative. Although they can account for non-linear interactions in the data and are scalable [7], Neural nets are often referred to as 'black boxes' (i.e., lacking explainability) [14,15,20,21,42–46] due to the difficulties in determining precisely how and why different input variables were used to produce the outputs.

Because we sought not merely to predict outcomes, but also to understand the relationships between multiple clinical variables and outcomes, we selected an 'explainable' AI solution, rather than a black box approach. Probabilistic Graphical Model-based [23–25,46] multimorbidity networks offer best-practice solutions to this problem. Moreover, they effectively model data without recourse to a fixed decision protocol (e.g decision trees), and are resilient to missing/unknown data. Crucially, the contributions of different combinations of variables to an outcome can be precisely and easily determined.

Explainability comes at a cost; unlike Neural nets, which are incredibly scalable, multimorbidity networks can model a maximum of only 30 or so variables at once [28,37,38]. It is therefore necessary to pre-identify high impact variables when modeling an outcome, a need fulfilled by PBC [10]. We argue that the ability to rigorously investigate interrelations among 30 or so primary determinants represents a giant step toward understanding cardiovascular disease.

Our results illustrate how multimorbidity networks provide explainable solutions for understanding the joint impacts of diagnoses, medications, and medical procedures on cardiovascular health outcomes. We emphasize that the necessarily brief results reported here hardly exhaust the contents of these machineries. Consider that a multimorbidity network with $n$ nodes supports $\sim 3^n$ possible queries. The net shown in **Fig 4B**, for example, supports $\sim 3^{14}$ different queries—a number that gives some indication both of the complexity of the data being extracted from the EHR corpus by our approach, and the value of these multimorbidity networks to further outcomes research.

## Conclusion

The analyses presented here provide a first step toward a global description of heart disease and associated comorbidities across the USA intermountain west. However, the map we seek resides not so much in the results reported here, as it does in the products of our analyses: the PGM multimorbidity networks. As we have explained, these networks support multitudes of queries, and when used in combination, support both wide-ranging and focused explorations of a disease landscape. Given the right datasets, we have shown that the approach can provide new insights, such as the mother-child cross-generational cardiovascular multimorbidities we described. However, our approach also has limitations. Our exact approach allows us to model at most $\sim 30$ health conditions at a time. In future work we would like to relax this limiting factor by allowing approximate solutions that enable us to scale up the complexity of the multimorbidity networks to thousands of health conditions. Another area for innovation regards incorporation of continuous variables, as current software packages do not allow us to incorporate such variables at scale, however there is no theoretical limitation preventing their use in a PGM framework.

A major strength of our approach is that these outcomes machineries can be redistributed as web-based tools. Indeed, the multimorbidity Networks described here have been made available online [pbc.genetics.utah.edu/lemmon2021/bayes], with the hope that the wider scientific community will find them useful for their own outcomes research. The ability to transform enormous collections of EHR data into compact, portable machines for outcomes

research, with no exchange of PHI, solves many of the legal, technological, and data-scientific challenges associated with large-scale EHR analyses.

## Supporting information

**S1 Fig. Distribution density plot of mother's age at pregnancy, with and without hypertension complicating pregnancy.** Blue line: mothers with diagnosis of hypertension complicating pregnancy (N = 11,523 mothers). Red line: mothers without diagnosis of hypertension complicating pregnancy (N = 113,491 mothers).
(TIF)

**S1 Table. Overview of Utah Data Resource.**
(TIF)

**S2 Table. Demographic variables and the Utah Data Resource.**
(TIF)

**S3 Table. Multimorbidity Landscape of Sinoatrial Node Dysfunction (SND) in adults.** Risk and fold-change risk estimates calculated from the multimorbidity network in **Fig 4B** main text. For detailed description of the clinical variables, please refer to **S5 Table**.
(TIF)

**S4 Table. Risks of Cardiac or Nervous System Congenital Anomalies as a Function of Comorbid Clinical Variables.** Risk of cardiac (**Panel A**) or nervous system (**Panel B**) congenital anomalies given the presence of specific clinical variables. Baseline risk and fold change risk calculated using the multimorbidity network in Fig 5 of the main text. For example, a child with a known diagnosis of Down Syndrome has a 25.9-fold increased risk of a cardiac congenital anomaly over the marginal risk of cardiac anomaly. HTN-PREG, hypertension complicating pregnancy (AKA pregnancy-induced hypertension). For detailed description of the clinical variables please refer to **S5 Table**.
(TIF)

**S5 Table. Reference table for EHR coding.**
(TIF)

## Acknowledgments

We thank Barry Moore, Jacob Shreiber, Jerry Rudisin, Sepideh Ebadi, Edward B. Clark and members of the University of Utah EDW, UPDB and Utah Center for High Performance Computing for insightful discussions, facilitating access to medical records and familial relationships, and computational support.

## Author Contributions

**Conceptualization:** Sergiusz Wesołowski, Gordon Lemmon, Thomas A. Miller, Bruce E. Bray, Vikrant G. Deshmukh, H. Joseph Yost, Karen Eilbeck, Martin Tristani-Firouzi, Mark Yandell.

**Data curation:** Sergiusz Wesołowski, Gordon Lemmon, Edgar J. Hernandez, Alex Henrie, Thomas A. Miller, Michael D. Puchalski, Bruce E. Bray, Rashmee U. Shah, Vikrant G. Deshmukh, Martin Tristani-Firouzi.

**Formal analysis:** Sergiusz Wesołowski, Gordon Lemmon, Edgar J. Hernandez, Alex Henrie, Bruce E. Bray, Martin Tristani-Firouzi, Mark Yandell.

**Funding acquisition:** H. Joseph Yost, Karen Eilbeck, Martin Tristani-Firouzi, Mark Yandell.

**Investigation:** Sergiusz Wesołowski, Gordon Lemmon, Thomas A. Miller, Derek Weyhrauch, Bruce E. Bray, Vikrant G. Deshmukh, Martin Tristani-Firouzi.

**Methodology:** Sergiusz Wesołowski, Gordon Lemmon, Edgar J. Hernandez, Vikrant G. Deshmukh, Mark Yandell.

**Project administration:** Michael D. Puchalski, Vikrant G. Deshmukh, H. Joseph Yost, Karen Eilbeck, Mark Yandell.

**Resources:** Michael D. Puchalski.

**Software:** Sergiusz Wesołowski, Gordon Lemmon, Edgar J. Hernandez, Alex Henrie, Mark Yandell.

**Supervision:** H. Joseph Yost, Martin Tristani-Firouzi, Mark Yandell.

**Validation:** Sergiusz Wesołowski, Gordon Lemmon, Alex Henrie, Thomas A. Miller, Derek Weyhrauch, Bruce E. Bray, Rashmee U. Shah, Vikrant G. Deshmukh, Rebecca Delaney, H. Joseph Yost, Karen Eilbeck.

**Visualization:** Sergiusz Wesołowski, Gordon Lemmon, Edgar J. Hernandez, Alex Henrie, Derek Weyhrauch, Bruce E. Bray, Rashmee U. Shah, Karen Eilbeck, Martin Tristani-Firouzi, Mark Yandell.

**Writing – original draft:** Sergiusz Wesołowski, Gordon Lemmon, Edgar J. Hernandez, Thomas A. Miller, Derek Weyhrauch, Michael D. Puchalski, Bruce E. Bray, Rashmee U. Shah, Rebecca Delaney, H. Joseph Yost, Karen Eilbeck, Martin Tristani-Firouzi, Mark Yandell.

**Writing – review & editing:** Sergiusz Wesołowski.

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
