## [Decision Letter · Decision Letter 0]

11 Oct 2021

PDIG-D-21-00066

An Explainable Artificial Intelligence Approach for Predicting Cardiovascular Outcomes using Electronic Health Records

PLOS Digital Health

Dear Dr. Wesolowski,

Thank you for submitting your manuscript to PLOS Digital Health. After careful consideration, we feel that it has merit but does not fully meet PLOS Digital Health’s publication criteria as it currently stands. Therefore, we invite you to submit a revised version of the manuscript that addresses the points raised during the review process.

We look forward to receiving your revised manuscript.

Kind regards,

Mecit Can Emre Simsekler, Ph.D.

Academic Editor

PLOS Digital Health

Journal Requirements:

1. We ask that a manuscript source file is provided at Revision. Please upload your manuscript file as a .doc, .docx, .rtf or .tex. If you are providing a .tex file, please upload it under the item type ‘LaTeX Source File’ and leave your .pdf version as the item type ‘Manuscript’.

2. Please provide separate figure files in .tif or .eps format only, and remove any figures embedded in your manuscript file. If you are using LaTeX, you do not need to remove embedded figures.

For more information about figure files please see our guidelines: https://journals.plos.org/digitalhealth/s/figures

3. We have noticed that you have uploaded supporting information but you have not included a list of legends. Please add a full list of legends for all supporting information files (including figures, table and data files) after the references list.

Additional Editor Comments (if provided):

For the general readership of the journal, it would be better if you could re-organize the section headings in a typical order, e.g., introduction, methods, results, discussion and conclusions. Accordingly, that would be great if you could add a short conclusion section highlighting the limitations of the study and directions for future research.

Please also ensure that the link on ‘data availability’ section you provided is accessible.

https://pbc.genetics.utah.edu/lemmon2021/bayes/bayes

Reviewers' comments:

Reviewer's Responses to Questions

**Comments to the Author**

1. Does this manuscript meet PLOS Digital Health’s publication criteria? Is the manuscript technically sound, and do the data support the conclusions? The manuscript must describe methodologically and ethically rigorous research with conclusions that are appropriately drawn based on the data presented.

Reviewer #1: Yes

Reviewer #2: Yes

2. Has the statistical analysis been performed appropriately and rigorously?

Reviewer #1: I don't know

Reviewer #2: Yes

3. Have the authors made all data underlying the findings in their manuscript fully available (please refer to the Data Availability Statement at the start of the manuscript PDF file)?

Reviewer #1: Yes

Reviewer #2: Yes

4. Is the manuscript presented in an intelligible fashion and written in standard English?

Reviewer #1: Yes

Reviewer #2: Yes

5. Review Comments to the Author

Reviewer #1: The authors used a comorbidity discovery method to analyse EHRs for comorbid diagnoses, procedures and medications. I believe that the paper fits well with the journal’s scope. I have a few suggestions for improving the understandability of the manuscript for a broader range of readers.

I understand that some of the authors developed the method, and the manuscript refers to the author’s previous paper where they explain the PBC. Yet, still, I would advise authors to provide more information on the method used.

I see that the authors highlighted some of the pros of using PBC, but could you please expand it? Why PBC? For instance, can’t we discover temporal relationships and quantify transition rates between comorbidities using other methods? What does PBC add to other methods?

Could you please check the links provided in the manuscript?; the links do not work.

Did your study provide some significant findings that have not been discovered yet? What does it add to the current knowledge on cardiovascular comorbidities? Would you please highlight accordingly? How can these findings be useful in practice?

Could you please check Figure 4B? Some texts are overlapped.

The authors might also suggest some directions for future research.

Reviewer #2: Overall this is a throughout and interesting approach to identifying how multiple conditionally dependent variables can predict certain cardiovascular outcomes using PBC. I especially found Table 1 to drive the central point that PBC can be a powerful statistical tool in smaller sample sizes as compared with chi-squared analyses for the same variables. The other illustrations are easy to follow and support the central arguments made in the text. Overall the case is well made for the utility of PBC derived multi morbidity networks in analyzing large EHR datasets.

There is an issue with figure 4B. I believe it is a duplicated image of figure 5A. The description of Figure 4B is totally dissimilar from what is depicted in the graphic. Please consider editing figure 4B prior to publication.

6. PLOS authors have the option to publish the peer review history of their article (what does this mean?). If published, this will include your full peer review and any attached files.

**Do you want your identity to be public for this peer review?** For information about this choice, including consent withdrawal, please see our Privacy Policy.

Reviewer #1: No

Reviewer #2: No

---

## [Editor Report · Decision Letter 1]

17 Nov 2021

An Explainable Artificial Intelligence Approach for Predicting Cardiovascular Outcomes using Electronic Health Records

PDIG-D-21-00066R1

Dear Dr. Wesolowski,

We're pleased to inform you that your manuscript has been judged scientifically suitable for publication and will be formally accepted for publication once it meets all outstanding technical requirements.

Within one week, you'll receive an e-mail detailing the required amendments. When these have been addressed, you'll receive a formal acceptance letter and your manuscript will be scheduled for publication.

An invoice for payment will follow shortly after the formal acceptance. To ensure an efficient process, please log into Editorial Manager at https://www.editorialmanager.com/pdig/ click the 'Update My Information' link at the top of the page, and double check that your user information is up-to-date. If you have any billing related questions, please contact our Author Billing department directly at authorbilling@plos.org.

Kind regards,

Mecit Can Emre Simsekler, Ph.D.

Academic Editor

PLOS Digital Health